# QUANTIFYING HUMAN-AI SYNERGY

## ABSTRACT

We introduce a novel Bayesian Item Response Theory framework to quantify human–AI synergy, separating individual and collaborative ability while controlling for task difficulty in interactive settings. Unlike standard static benchmarks, our approach models human–AI performance as a joint process, capturing both user-specific factors and moment-to-moment fluctuations. We validate the framework by applying it to human–AI benchmark data (n=667) and find significant synergy. We demonstrate that collaboration ability is distinct from individual problem-solving ability. Users better able to infer and adapt to others' perspectives achieve superior collaborative performance with AI—but not when working alone. Moreover, moment-to-moment fluctuations in perspective taking influence AI response quality, highlighting the role of dynamic user factors in collaboration. By introducing a principled framework to analyze data from human-AI collaboration, interactive benchmarks can better complement current single-task benchmarks and crowd-assessment methods. This work informs the design and training of language models that transcend static prompt benchmarks to achieve adaptive, socially aware collaboration with diverse and dynamic human partners.

## 1 INTRODUCTION

Benchmarks have played a pivotal role in advancing large language model (LLM) development. Popular examples such as BIG-Bench (Srivastava et al., 2023), MMLU (Hendrycks et al., 2021), ARC (Clark et al., 2018), and GSM8K (Cobbe et al., 2021) evaluate LLM performance on fully specified static prompts (non-interactive prompting), where models solve well-defined problems independently (Glazer et al., 2024; Rein et al., 2024; Phan et al., 2025; Jimenez et al., 2023).[1] While such benchmarks have produced models that excel at solving closed, curated problems (Luong & Lockhart, 2025), optimizing for these benchmarks has contributed to three widely recognized issues: (1) reduced effectiveness on complex, real-world tasks (Becker et al., 2025; Mancoridis et al., 2025); (2) limited collaborative abilities, often manifesting as "sycophantic" behavior rather than genuine assistance, and communication breakdowns (Shojaee et al., 2025; Perez et al., 2023; Bansal et al., 2024); and (3) a focus on imitating human capabilities rather than complementing or extending them (Srivastava et al., 2023; Haupt & Brynjolfsson, 2025; Mancoridis et al., 2025; Acemoglu & Restrepo, 2018; Riedl, 2024; Shao et al., 2025).

To complement existing AI benchmarks, some recent efforts have focused on human-AI teams, assessing model performance in situations that more closely resemble real-world deployment (Haupt & Brynjolfsson, 2025; Shao et al., 2024; Doshi & Hauser, 2024). This approach is motivated by the insight that intelligence—whether human or artificial—is inherently interactive, contextual, and collaborative (Minsky, 1986; Hutchins, 1995; Lévy, 1997; Riedl et al., 2021). Sophisticated thinking rarely occurs in isolation; it emerges instead through dialogue, feedback, refinement, and the integration of diverse perspectives. One example of such a benchmark is Chang et al. (2025). They adapt a well-known benchmark (MMLU) to provide evidence about the average performance of hu-

---

[1]While our focus is on LLMs, we sometimes refer to these systems as "AI models" or simply "AI," acknowledging that LLMs represent only a subset of artificial intelligence. We also acknowledge that that there is debate whether LLMs are really intelligent, even though "AI" implies it.

mans working alone, AI alone, and human-AI pairs. Yet benchmarks alone are insufficient—they must be paired with methods that rigorously quantify the synergy arising from these interactions.[2]

Prior work has observed substantial variation in AI's impact depending on task type, task difficulty, and user skill level (Dell'Acqua et al., 2023; Becker et al., 2025; Vaccaro et al., 2024), sometimes reporting conflicting results about who benefits and on which tasks (Brynjolfsson et al., 2025; Noy & Zhang, 2023; Becker et al., 2025; Choi et al., 2024; Caplin et al., forthcoming; Vaccaro et al., 2024). For example, Otis et al. (2024) and Riedl & Bogert (2024) show that higher-skilled users benefit more, whereas others find that lower-skilled users benefit more (e.g., Brynjolfsson et al., 2025; Noy & Zhang, 2023), or find few benefits at all (Vaccaro et al., 2024). Prior work has also typically focused on aggregate comparisons across experimental conditions, which makes it difficult to explain the differential impact of AI assistance for different users on different tasks (e.g., Chang et al., 2025). Understanding heterogeneous impacts better would help inform model development and training.

To better explain AI's impact, we draw on established theories from human-human collaboration, particularly Theory of Mind (ToM). ToM refers to the capacity to represent and reason about others' mental states (Premack & Woodruff, 1978). It plays a crucial role in human interaction (Nickerson, 1999; Lewis, 2003), allowing individuals to anticipate actions, disambiguate and repair communication, and coordinate contributions during joint tasks (Frith & Frith, 2006; Clark, 1996; Sebanz et al., 2006; Tomasello, 2010). ToM has repeatedly been shown to predict collaborative success in human teams (Weidmann & Deming, 2021; Woolley et al., 2010; Riedl et al., 2021). Its importance is also recognized in AI and LLM research (Prakash et al., 2025; Liu et al., 2025), for purposes such as inferring missing knowledge (Bortoletto et al., 2024), aligning common ground (Qiu et al., 2024), and cognitive modeling (Westby & Riedl, 2023). We argue that working with LLMs similarly requires adaptive, interaction-sensitive behavior due to the dialogic nature of human-AI interactions and the semi-autonomous, unpredictable nature of LLM output (see Appendix for theoretical justification for studying ToM in human-AI interactions).[3]

In this paper, we build on the recent emergence of human-AI benchmarks by proposing and validating a principled framework for analyzing human-AI interaction data to quantify and explain human-AI synergy. Viewing these interactions through the lens of teamwork, we apply Item Response Theory and Bayes shrinkage to account for differences in task difficulty and user ability. Crucially, our framework distinguishes between and estimates each user's "individual ability" ($\theta$) and "collaborative ability" ($\kappa$, see Methods). We validate this approach using data from Chang et al. (2025) in which 667 humans completed tasks with and without AI assistance across math, physics, and moral reasoning. We benchmark two AI models of different capacity and capability—GPT-4o and Llama-3.1-8B—and quantify the extent to which they improve human performance, controlling for variation in user ability and task difficulty. Finally, we examine which users benefit most from AI collaboration and investigate why—focusing on the role of Theory of Mind.

Our paper makes two key contributions. First, we introduce and empirically validate a framework for benchmarking human–AI synergy—a new paradigm for evaluating LLMs that goes beyond static, single-task accuracy metrics and crowd-judgment scoring. Our approach quantifies how much different AI models improve user performance over solo baselines, while controlling for task difficulty and user ability. In doing so, it estimates user-specific performance gains from AI collaboration and separately identifies each user's individual and collaborative ability. This enables fine-grained comparisons of models' capacity to enhance the collective intelligence of human–AI teams in realistic problem-solving settings. Second, we identify Theory of Mind as a key cognitive mechanism in human–AI synergy. Users with stronger ToM achieve superior collaborative performance with AI—but not when working alone—and both stable individual differences and moment-to-moment fluctuations in ToM predict AI response quality. These findings suggest that ToM-like capabilities—and the ability to adapt to users' social-cognitive states—are critical for LLMs intended for interactive, dynamic problem-solving, and point to a new research and development agenda for building socially

---

[2]We define "synergy" simply as an uplift in human performance when given access to an LLM system (see Methods for details).

[3]Early empirical work found that users benefit from forming accurate mental models of AI Bansal et al. (2019b); Paleja et al. (2021); Bansal et al. (2019a); Gero et al. (2020); Alipour et al. (2021), though this effect is not always consistent (Fügener et al., 2022).

aware, adaptive AI partners. Together, these contributions open a path toward designing AI systems that prioritize emergent human-AI synergy over standalone performance.

## 2 METHODS

### STUDY DESIGN AND DATA

Our data comes from the ChatBench assessment constructed by Chang et al. (2025). ChatBench takes the popular MMLU benchmark (Hendrycks et al., 2021) and adapts it into a human-AI benchmark with three components: i) solo AI in which AI models problems alone; ii) solo human, in which humans solve problems alone; iii) human-AI pairs, in which a human partners with an AI model to solve problems. ChatBench contains multiple-choice questions on three domains: mathematics, physics, and moral reasoning. The dataset includes a total of 396 questions covering a wide range of difficulties. Questions were randomly assigned to users and shared across solo and joint phases, ensuring direct comparability. On average, each question was answered by about 20 different users in total—about 5 users in solo work and 15 jointly with AI—providing overlap that supports stable item-level estimates.

Overall, the data contains responses from 667 humans. Participants begin the study by answering three questions by themselves (solo human). For our purposes this step is crucial as it allows for the independent estimation of individual human ability (Weidmann & Deming, 2021). In Phase 2, users receive AI assistance via a chat interface and are randomly assigned to either GPT-4o or Llama-3.1-8B (this random assignment is fixed on the user-level and does not change by question; i.e., a user works with either GPT-4o or Llama-3.1-8B, but not both). In this second phase users answer nine more questions. When partnering with an AI, participants are required to type something in the AI chat interface in order to progress, but the nature of the interaction was left entirely up to the user. For more details, see Chang et al. (2025).[4]

### DECOMPOSING AI IMPACT ACROSS- AND WITHIN-USERS

We analyze the data using a form of multilevel modeling known as Item Response Theory (IRT, Baker & Kim, 2004) and a Bayesian workflow Gelman et al. (2020). A core strength of multilevel models is their ability to account for complex dependency structures (Gelman & Hill, 2007). In our case, the data has three main structural components: questions (items), users, and working solo or jointly with AI. Users respond to multiple questions and questions are answered by multiple users (with partial crossing such that different users answer different questions). IRT allows us to decompose the binary response of answering questions correctly (using the logistic item-response model) along two crucial dimensions using random intercepts and slopes. In general, the model takes the form:

$$\Pr(Y_{ij} = 1) = \text{logit}^{-1}(\text{Ability}_i - \text{Difficulty}_j)$$

where ability$_i$ represents the problem-solving capacity of user $i$ (either working solo or jointly with AI) and difficulty$_j$ represents the difficulty of question $j$.

Using this basic structure, we estimate the problem-solving capacity of users working alone ($\theta$) and working jointly with an AI model ($\kappa$). We index different AI models as $AI$:

$$\Pr(Y_{ij} = 1) = \begin{cases} \text{logit}^{-1}(\theta_i^{\text{human}} - \beta_j) & \text{if } joint_{ij} = 0 \text{ (solo)} \\ \text{logit}^{-1}(\underbrace{\kappa_{i,\text{AI}}^{\text{total}}}_{\text{Ability}} - \underbrace{(\beta_j + \gamma_j)}_{\text{Difficulty}}) & \text{if } joint_{ij} = 1 \text{ (with AI)}. \end{cases} \tag{1}$$

Here, $\theta_i^{\text{human}}$ is user $i$'s latent ability when working alone, estimated from responses to items in the solo condition. $\kappa_{i,\text{AI}}^{\text{total}}$ represents the overall performance of user $i$ working with AI. Question difficulty is decomposed into the question's solo difficulty ($\beta_j$) and how difficult it is to collaboratively

---

[4]In Phase 2, participants were also randomly assigned to two different modes of AI interaction: "answer first" in which humans provided a provisional answer before gaining access to AI assistance, and "direct-to-AI" in which participants have immediate access to AI help. In our analysis, we either use only data from the direct-to-AI condition or control for these different conditions using fixed effects. Note also that users had a direct piece-rate financial performance incentive to answer questions correctly ($0.10 per correct answer).

solve the question ($\gamma_j$). $\kappa^{\text{total}}_{i,\text{AI}}$ estimates the strength of the human-AI pair which includes contributions from the human's solo ability as well as the collaborative ability of the human and the AI. Importantly, this model allows us to estimate the *boost* that each individual receives from working with AI. This requires a second analytical step. Specifically, we define *boost* as $\kappa^{\text{total}}_{i,\text{AI}} - \theta^{\text{human}}_i$ (all based on estimates from Eq. 1). For example, say users $A$ and $B$ have the same solo ability ($\theta^{\text{human}}_A = \theta^{\text{human}}_B$) but that $A$ consistently scores higher when working jointly with the same AI as user B. In this case user $A$ has a larger value of $\kappa^{\text{total}}_{A,\text{AI}}$ than user $B$ (i.e. $\kappa^{\text{total}}_{A,\text{AI}} > \kappa^{\text{total}}_{B,\text{AI}}$). Consequently, user $A$'s AI boost ($\kappa^{\text{total}}_{A,\text{AI}} - \theta^{\text{human}}_A$) will be bigger than user $B$'s ($\kappa^{\text{total}}_{B,\text{AI}} - \theta^{\text{human}}_B$).

Estimating user-specific AI boost helps answer the question of *who benefits most from partnering with AI?* This helps to build an understanding of the evolving dynamics of human capital, inequality, and productivity in an AI-augmented economy. Moreover, it allows researchers to explain why some people benefit more than others, informing the debate on whether AI acts as a complement or substitute to human labor, which may help in informing policies for AI deployment. Conceptually, "AI boost" reflects both emergent synergy—gains arising from interaction, clarification, scaffolding, or co-construction—and direct the AI's contributions, such as cases where the AI provides the correct answer and the user adopts it. The boost should therefore be interpreted as a behavioral measure of the emergent human-AI collaborative performance, including the full range of mechanisms through which human-AI collaboration may affect performance.

In the model from Eq. 2, we further decompose joint performance into the collaborative ability of the user and the AI model. This approach can be used to *benchmark* different models. To illustrate this, different AI models are indexed as $AI_m$:[5]

$$\text{Pr}(Y_{ij} = 1) = \begin{cases} \text{logit}^{-1}(\theta^{\text{human}}_i - \beta_j) & \text{if } joint_{ij} = 0 \text{ (solo)} \\ \text{logit}^{-1}(\underbrace{\kappa^{\text{human}}_{\text{i}} + \kappa^{\text{AI}}_m}_{\text{Ability}}) - \underbrace{(\beta_j + \gamma_j)}_{\text{Difficulty}}) & \text{if } joint_{ij} = 1 \text{ (with } AI_m). \end{cases} \quad (2)$$

We can interpret $\kappa^{\text{human}}_i$ as the collaborative ability of user $i$ (when working jointly with AI). This collaborative ability includes the user's ability in deciding when and how to delegate tasks to the AI assistant, how to formulate the delegation (i.e., writing LLM prompts), deciding whether to accept the AI's response, providing refinements or requesting clarification, and so forth. Similarly, $\kappa^{\text{AI}}_m$ represents the collaborative capability of the AI model (that user $i$ is paired with). This parameter is an important object for AI benchmarking purposes as it quantifies the extent to which different models amplify human performance. Using ChatBench data, for example, we are able to compare $\kappa^{\text{AI}}_{\text{GPT4o}}$ and $\kappa^{\text{AI}}_{\text{Llama}}$ (see Figure 1B).

Overall, the model in Eq. 2 is well suited to AI benchmarking as it specifically estimates the capability of each AI model to raise the performance of the average user, controlling for differences in user solo ability $\theta^{\text{human}}_i$, user collaborative ability ($\kappa^{\text{human}}_{\text{i}}$), and question difficulty ($\beta_j$ and $\gamma_j$).

Throughout these analyses, models are fit using Bayesian inference via the `brms` package in `R` (Bürkner, 2017). Estimates are based on Bayesian shrinkage which helps improve generalization and prevents overfitting by stabilizing estimates, especially when data are limited or unevenly distributed across units (e.g., users or items; Gelman & Hill, 2007). All reported results are from converged models with $\hat{R}$ values in the commonly accepted range (Bürkner, 2017). To explore the specific role of ToM we also estimate a third model (see Appendix).

Our analytical framework relies on several assumptions and data requirements. First, the Item Response Theory structure assumes monotonicity and unidimensionality. We assume respondents have a single latent ability when they're working alone, or working with AI — and that the probability of correct answers monotonically increases with ability. Second, to identify and estimate solo and collaborative ability our setup requires item crossing: i.e. that the same questions are tackled by participants working alone, and working with AI. The model then assumes that the decomposition of difficulty is additive, i.e., that collaborative difficulty supplements (but does not qualitatively alter) the underlying difficulty of each item. Third, our framework tests—rather than assumes–whether joint human–AI performance can be decomposed into additive contributions of the human and the

---

[5]The $m$ subscripts can also be applied to Eq. 1, but we suppressed these for clarity of exposition.

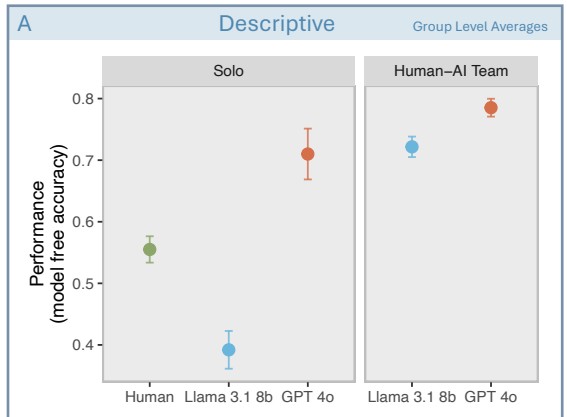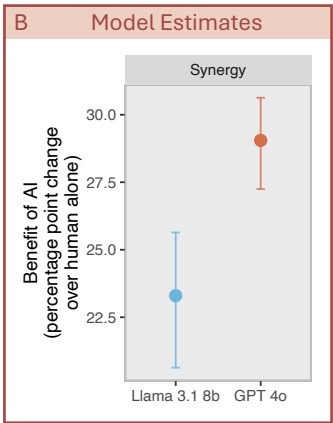

Figure 1: **AI improves performance in human-AI teams. A** Model-free analysis of group-level averages (AI-alone accuracy is few-shot letter-only). Human-AI teams perform substantially better than human-alone, even with the weaker Llama-3.1-8B model. **B** The "AI Synergy" benchmark. Average synergy with GPT-4o is significantly higher than Llama-3.1-8B. (IRT estimates based on Eq. 2 accounting for task difficulty and user ability; bars are 95% CIs).

AI model. We treat human collaborative ability and model-specific collaborative capability as separable components of team performance. This implies that interactions between particular users and particular models do not introduce systematic idiosyncratic effects beyond those captured by random variation. Finally, the Bayesian hierarchical structure assumes appropriate partial pooling across users and items, enabling stable estimation under sparse or uneven data. This approach presumes that user- and item-level effects are exchangeable within their respective distributions. Together, these assumptions allow us to interpret our estimates of solo ability, collaborative ability, task difficulty, and AI-induced performance gains as reflecting stable and separable components of human–AI collaboration.

## 3 RESULTS

**Quantifying Synergy.** We begin with a model-free analysis (Figure 1A). Working alone, humans answer 55.5% of questions correctly on average (based on 2,072 solo observations). Working alone, GPT-4o performs better than the human average (71% accuracy) whereas Llama-3.1-8B does worse (39%).[6] When working jointly with humans, even the relatively weak Llama-3.1-8B model achieves substantially higher performance, significantly outperforming human-alone performance. GPT-4o-with-human also achieves substantial gains, outperforming GPT-4o-alone. However, the large difference in solo performance between Llama-3.1-8B and GPT-4o shrinks dramatically. We provide additional details on the mechanism of complementarity, including a break down by task difficulty and domain in Appendix Section H.

Next, we benchmark the collaborative capability of each AI model by quantifying its synergy (Figure 1B). We extend the results of Chang et al. (2025) by quantifying the average impact each model has on human-AI performance *while accounting for differences in task difficulty, human solo ability, and human collaborative ability*. We find that Llama-3.1-8B boosts performance by 23 percentage points on average, while GPT-4o boosts performance by 29 percentage points. In addition to these point estimates, our approach provides uncertainty estimates about the extent to which different models boost performance of the average human user. We find no overlap in the confidence intervals, suggesting GPT-4o has a clear advantage in terms of its collaborative capability.[7]

---

[6]This performance difference is not surprising given that GPT-4o is a much larger model.

[7]As a robustness test (see Appendix), we estimate a three-phase model that uses an additional 3,056 observations from a a second "solo first" condition and separately identifies learning or fatigue effects from working with AI, confirming that the main results are not driven by the fact that the with-AI condition always followed the solo condition and, if anything, slightly underestimate AI's true benefit.

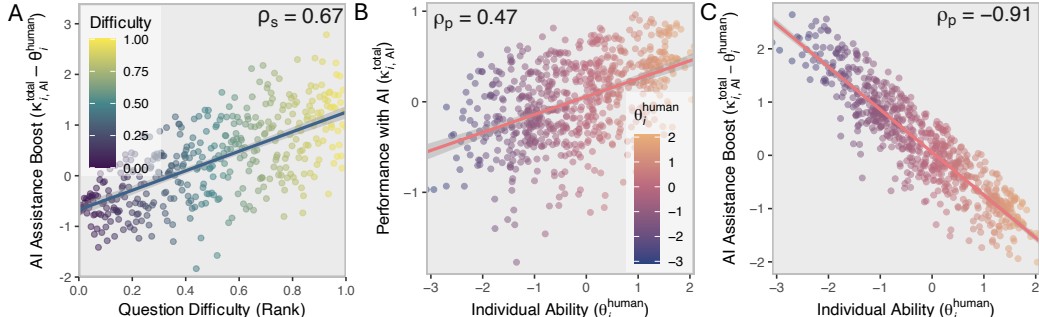

Figure 2: **Who benefits from AI and on which task?** **A** Highest AI boost is achieved on the most difficult tasks. Points represents questions. **B** Higher-ability users typically perform at a higher absolute level when working with AI assistance. Point represent users. **C** Absolute AI Boost is highest for lower-ability users. Note that the AI boost is limited by ceiling effect so that individuals with higher ability receive a relatively lower boost from working with AI. Lines are best-fit linear regressions. Point represent users. All results based on IRT estimates.

**Within-User Ability Decomposition.** To test whether working alone and with AI are separately estimable latent abilities we fit the model from Eq. 1 (see Methods) and a nested, simpler model that estimates a single latent ability for each user. We perform a Bayesian model comparison with leave-one-out cross-validation (LOO) which evaluates how well each model predicts unseen data by estimating the expected log predictive density (ELPD) for held-out observations. We find the full model fits the data substantially better ($\Delta$ELPD = 50.9, SE = 10.2). Overall model convergence is excellent (all $\hat{R}$ are 1.00 showing no evidence of chains mixing poorly) and both Bulk ESS and Tail ESS are large (all $> 1,500$, some $> 4,000$), indicating very good sampling efficiency and stable posterior estimates. This provides strong evidence for condition-specific latent abilities—one reflecting users' ability to work alone ($\theta$), and one capturing their ability to work with AI ($\kappa$)—supporting the theory that $\theta \neq \kappa$. These estimates are based on responses to multiple questions within a single topic domain (physics, math, or moral reasoning). While we interpret $\theta$ and $\kappa$ as reflecting true latent abilities, they are domain-bound and should not be interpreted as general-purpose skills. In this sense, our estimates reflect context-specific expressions of ability rather than fully general traits.

**Who benefits from AI?** Having quantified the average synergy for each model—Llama-3.1-8B and GPT-4o—we now focus on decomposing and understanding these effects. First, we estimate and analyze individual-level AI effects *across* users. We then identify individual ability and collaborative ability *within* each user. As individual users work on problems with different difficulty levels, to estimate the impact of AI across users we must control for differences in task difficulty (using the IRT approach outlined above). As part of this analysis we find that AI benefits are not uniform across task difficulty (Figure 2A). Specifically, the more difficult tasks are for humans working alone, the more they benefit from AI. This is consistent with theories of human–machine complementarity suggesting that AI acts as a cognitive amplifier in these situations (Autor, 2015; Acemoglu & Restrepo, 2018).

Having controlled for differences in task difficulty, we are able to estimate user-level AI boost (i.e., the difference for each user between their solo and with-AI performance, controlling for task difficulty). Specifically, we test whether lower- or higher-ability users benefit more from AI (Figure 2B). We find a strong positive correlation between individual solo performance ($\theta_i^{human}$) and team performance with AI ($\kappa_{i,AI}^{total}$). In brief, the highest-ability users are still the best performers when working with AI (Figure 2B), again supporting the theories of human–machine complementarity. However, the AI boost estimates suggest that lower-ability users gain more than their higher-ability peers (Figure 2C). This result is consistent with prior studies demonstrating that for closed-form tasks where humans can rely on AI answers, AI tends to reduce output inequality and helps level the

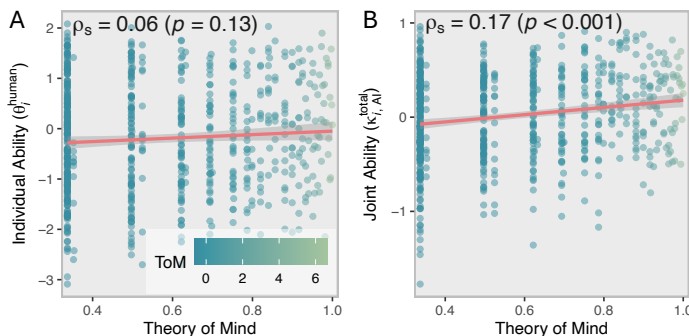

Figure 3: **Theory of Mind differentially predicts performance during collaboration with AI but not when working alone.** **A** ToM is not significantly correlated with individual solo ability. **B** ToM is significantly correlated with the AI collaboration ability. Ability are IRT estimates from Eq. 1 ToM is raw data aggregated to the user-level. Lines are best-fit linear regressions.

playing field (e.g., Noy & Zhang, 2023; Dell'Acqua et al., 2023).[8] That is, we find evidence of both skill complementarity *and* equalizing effects. However, the equalizing effect is not strong enough to fully offset the initial differences and complementarity effect. We also test whether GPT-4o or Llama-3.1-8B is better at complementing users at different ability levels. For example, higher-ability users might benefit more from stronger models due to their higher capacity to evaluate and leverage suggestions. Conversely, lower-ability users might benefit more from stronger models due to better scaffolding and structured thinking support that such models might provide. We reject the hypothesis of heterogeneous effects: the impact of both models is stable across the distribution of solo-ability ($\kappa_{\text{Llama}}^{\text{AI}}$ 0.02 [95% CI: -0.12, 0.17] compared to $\kappa_{\text{GPT4o}}^{\text{AI}}$ baseline).

**Theory of Mind.** What explains the observed human-AI synergy? We use our analytical framework to help make progress on this question by examining the role played by Theory of Mind in human-AI interaction. We collect task-level Theory of Mind measures from the users' portion of the dialogue—exclusive of the AI assistant responses—using a language model as a research assistant (LMRA, Eloundou et al., 2024, see Appendix). We followed best practices (Li et al., 2025) and performed extensive robustness test to validate construct validity including (a) collecting data from a variety of leading models, (b) explored robustness to the specific instruction set in the prompt, (c) order bias in the prompt, and (d) absolute vs. relative judgments. We find excellent inter-coder reliability among the different LLM "raters". We also collected gold standard human ratings for a stratified random sample of 120 dialogues from 32 individuals to validate the approach of studying dialogue-level ToM signatures in human–AI collaboration and the LMRA assessment (see Appendix). First, we find that ToM differentially predicts performance in the AI condition, but not in the solo condition. We find that ToM is a significant positive moderator for collaborative ability (Model from Eq. 3: 0.65 [95% CI: 0.01, 1.29]), but *not* solo ability (0.41 [95% CI: -0.17, 0.99]).[9] Similarly, we find that ToM positively predicts collaborative ability with AI (Figure3A) but does not predict individual solo ability (Figure 3B). Taken together, these findings support the idea that Theory of Mind—belief tracking, perspective taking, goal inference etc.—captures a user-specific form of social-cognitive skill which is important in understanding human-AI synergy.

The above analysis raises an interesting question: is ToM a fixed user-level trait that is merely associated with an individual's latent ability to collaborate with AI—or can we view ToM as a direct mechanism through which collaboration synergy unfolds? These are testable questions in that a direct relationship implies that a user's ToM would influence the quality of human-AI dialogues and, more specifically, the quality of the AI responses. Moreover, a mechanistic relationship leaves

---

[8]This result is partly driven by the fact that high-ability users have less room for improvement (i.e., a ceiling effect).

[9]We find substantively similar results using a different LLM to assess ToM signatures in users' prompts. Specifically, using Google's Gemini 2.5 Pro we find significant ToM in the joint condition with 0.58 [95% CI: 0.03, 1.12]), but *not* solo ability (0.34 [95% CI: -0.16, 0.84]).

| Dependent Variable: | Quality of AI Responses | | | | |
|---|---|---|---|---|---|
| | OLS | | | | Bayesian |
| | (1) | (2) | (3) | (4) | (5) |
| User ToM (leave-one-out mean) | $0.16^{**}$ | $0.16^{**}$ | $0.25^{***}$ | $0.27^{***}$ | $0.31^{*}$ |
| | (0.05) | (0.05) | (0.05) | (0.05) | [ 0.23; 0.39] |
| Solo Ability | | 0.04 | 0.01 | 0.01 | $-0.00$ |
| | | (0.04) | (0.03) | (0.03) | [-0.06; 0.05] |
| Within-User ToM Deviation (plain) | | | | $0.09^{**}$ | $0.10^{*}$ |
| | | | | (0.04) | [ 0.05; 0.15] |
| Controls | | | | | |
|   Effort 1 (Turns) | | | $-0.15^{***}$ | $-0.17^{***}$ | $-0.16^{*}$ |
| | | | (0.03) | (0.04) | [-0.21; -0.12] |
|   Effort 2 (Total Chars) | | | $0.33^{***}$ | $0.32^{***}$ | $0.20^{*}$ |
| | | | (0.06) | (0.05) | [ 0.15; 0.25] |
|   Effort 3 (Edit Ratio) | | | $-0.77^{***}$ | $-0.78^{***}$ | $-0.78^{*}$ |
| | | | (0.04) | (0.04) | [ -0.74; -0.82] |
| Num. obs. | $5,868$ | $5,868$ | $5,868$ | $5,868$ | $5,868$ |
| Num. groups: Questions | 396 | 396 | 396 | 396 | 396 |
| Adj. R$^2$ | 0.23 | 0.40 | 0.40 | 0.40 | |

$^{***}p < 0.001; ^{**}p < 0.01; ^{*}p < 0.05$ (or Null hypothesis value outside the confidence interval).

Table 1: **Theory of Mind, both as user-level trait and within-user variation, cause higher quality AI responses.** OLS models show standard errors clustered by individual. Bayesian model shows 95% confidence intervals.

open the possibility that dynamic within-user changes in ToM expression could influence the quality of AI responses.

To explore these questions we examine the relationship between ToM and AI response quality (Table 1). We use a LMRA to assess the quality of AI responses (see Appendix). Using variance decomposition (see Methods) we find robust evidence that ToM as a fixed user-level trait is related to higher AI response quality (Model 1 $\beta = 0.16$, $p = 0.0013$). Solo ability does not lead to higher quality AI responses (Model 2: $\beta = 0.04$, $p = 0.34$) and including it as a control does not change the estimate for the ToM trait. This mirrors theoretical predictions from human teamwork which show that synergy in small-group settings is positively related to ToM, but largely unrelated to individual problem-solving ability (Weidmann & Deming, 2021).[10] The effect of ToM increases in size and significance when fine-grained controls for effort are included (Model 3: $\beta = 0.25$, $p < 0.001$) indicating that ToM reflects a deeper cognitive skill beyond motivation and effort. This suggests that ToM by itself—independent of engagement—affects AI response quality. Moreover, we find a significant positive effect of dynamic, with-question variation in ToM (Model 4: $\beta = 0.09$, $p = 0.007$) suggesting that ToM is not only a stable trait but varies situationally question-by-question. We find consistent, and slightly larger effects for both the ToM trait and ToM as mechanism using the hierarchical Bayesian model with confidence intervals excluding zero (Model 5).

In summary, we find trait-level ToM is a strong and consistent predictor of AI response quality. This suggests that ToM is a stable, trait-like ability that allows humans to reason about AI behavior. This shapes the effectiveness of human-AI collaboration similar to the way it shapes human-human interaction (Riedl et al., 2021). However, we find Theory of Mind is not only a stable trait but also varies dynamically. By decomposing Theory of Mind into a stable trait and a dynamic, question-specific mechanism, we move to a better understanding when and why human-AI synergy emerges. This is consistent with results showing that ToM it requires cognitive effort to deliberately take others perspective and can be hampered by cognitive load (Qureshi et al., 2010; Lin et al., 2010), and that it can be facilitated by context (Meijering et al., 2010).

## 4 RELATED WORK

Static benchmarks like MMLU (Hendrycks et al., 2021), BIG-Bench (Srivastava et al., 2023) and GSM8K (Cobbe et al., 2021) have driven and guided LLM development. However, researchers have noted that over-reliance on these benchmarks risks narrowing model development toward imitating

---

[10]Similarly, Markiewicz et al. (2024) report related findings that ToM predicts cooperation while intelligence alone does not.

human skills rather than complementing human capabilities (Srivastava et al., 2023; Acemoglu & Restrepo, 2018; Haupt & Brynjolfsson, 2025; Riedl, 2024). Recent empirical work has begun to focus on measuring human-AI collaborative outcomes, both in field settings (Brynjolfsson et al., 2025; Noy & Zhang, 2023) and in human-AI benchmarks (Chang et al., 2025). Analyzing benchmark data from interactive human-AI collaboration, however, is challenging. Lalor et al. (2016) is a notable work pioneering the use of Item Response Theory for the evaluation of NLP methods and some work has started exploring methods for interactive evaluation, e.g., in the medical domain (Liao et al., 2024), reasoning tasks (Huang et al., 2023), or multimodal settings (Raza et al., 2025). These studies, along with others in social science (Dell'Acqua et al., 2023; Riedl & Bogert, 2024, e.g.,), have provided initial evidence that the synergy between humans and AI is heterogeneous and depends on both the user and the task, prompting the need for better analysis frameworks to inform model development and training. We extend this empirical literature by providing new analytical tools to quantify and understand human-AI synergy. By modeling user- and task-level heterogeneity in the effects of AI assistance our work advances our understanding of complementarity, which tasks benefit disproportionately from AI assistance, and how AI assistance interacts with learning.

In parallel, building on insights from social science pointing to the important role of cognitive and social mechanisms underlying effective collaboration (Woolley et al., 2010; Weidmann & Deming, 2021), many papers in computer science shown that effective human-AI interaction also depends on the ability to form mental models (Bansal et al., 2019b; Paleja et al., 2021; Bansal et al., 2019a; Gero et al., 2020; Alipour et al., 2021; Kelly et al., 2023). Notably, both directions are important: AI forming mental models about the human as well as the human forming mental models about the AI (Fügener et al., 2022; Westby & Riedl, 2023). Recent work has benchmarked different aspects of ToM in LLMs (Kim et al., 2023; Shapira et al., 2023; Strachan et al., 2024; Shapira et al., 2024) and recent mechanisitc studies of LLMs have tried to better understand LLMs' ToM capacity Zhu et al. (2024); Prakash et al. (2025). Our work builds on these research strands by illustrating the importance of perspective-taking in human-AI synergy, and provides a framework how to measure both latent ToM reasoning as well as moment-to-moment fluctuations in human–AI collaborations.

## 5 DISCUSSION

The recent emergence of human-AI benchmarks (Chang et al., 2025; MindGames Arena Hub Organizers, 2025) has highlighted the need for evaluations that better reflect real-world deployment environments. Building on this, our approach advances the field by providing tools to quantify, benchmark, and design human-AI synergy. We introduce a novel two-stage Bayesian estimation framework based on Item Response Theory that offers a principled method for measuring the synergy between human users and LLMs. By estimating the impact of AI on individual users while controlling for differences in task difficulty, user ability, and shrinking noisy estimates, our framework offers a deeper understanding of human-AI collaboration: it allows us to quantify human–AI synergy as the incremental value AI provides to individual users. This also shows that some users benefit more from AI than others and points to the importance of considering the entire "synergy distribution" across ability levels as a metric. We also demonstrate that working with AI requires fundamentally different skills than working alone, which can be separately measured and quantified. While existing work describes performance differences between models and conditions (e.g., Chang et al., 2025) and focuses on stand-alone performance (e.g., Hendrycks et al., 2021), our setup provides explicit estimates of uncertainty and benchmarks AI model by quantifying the synergy they provide to human interaction.

By showing that AI performance can improve significantly through collaboration with humans, this research challenges the dominant model-centric paradigm of benchmarking AI systems solely on their standalone performance. Our measurement framework can serve as a diagnostic tool for AI design. The two-stage IRT approach isolates which cognitive or interactional skills are amplified by AI and which are left untouched. With this, designers can target models for specific complementary skills (e.g., inference, perspective-taking, working memory support) instead of treating general helpfulness as the only objective. This moves AI improvement from a black-box performance race toward an evidence-driven skill-matching approach.

The presence of strong human-AI synergy demonstrates that response quality is not an inherent property of the model alone but emerges from the interaction between human reasoning and AI

capabilities. Theory of Mind appears to play an important role in this regard and functions as a cognitive-behavioral interface that enables the emergence of this synergy through patterned interaction. User-level ToM strongly predicts AI complementarity, but not solo performance, suggesting that the ability to model another agent's mental states—whether human or artificial—fundamentally shapes collaborative outcomes. Further, we build on evidence that ToM is a dynamic construct that can vary within-users (Qureshi et al., 2010; Schneider et al., 2012; Lin et al., 2010) and find that within-user differences in ToM influence the quality of human-AI dialogue. This suggests that LLMs should not treat user profiles as static. Instead, prompt histories should be viewed as signal-rich, dynamic behaviors that need to be modeled over time. Future models could be built to better recognize and adapt to these shifts, explicitly supporting emergent synergy. Future work could explore whether variation in user's ToM level triggers or suppresses ToM-relevant circuits inside LLM (Prakash et al., 2025), which could inspire joint probes of both users and models. The insight that ToM varies dynamically supports the idea that users modulate their behavior—for example based on the affordance of the task—and can guide model training and interface design (Buckner, 2024).

By providing both methodological tools and an analysis of human-AI collaboration, this work lays a foundation for understanding and optimizing interactions that move beyond simple performance comparisons toward theoretical explanations of when, why, and how humans and AI systems can achieve synergistic outcomes. Our approach transforms human-AI collaboration from an anecdotal observation into a measurable, optimizable dimension of AI capability, enabling the design of systems explicitly aimed at emergent collaborative synergy rather than standalone performance. Our findings suggest that Theory of Mind is not merely a capability that allows LLMs to appear more human-like, but a mechanism that can transform human capability in collaborative interaction. By combining new benchmarks with the analytical tools introduced here, future research and development can systematically build and evaluate models designed to support emergent synergy and enhance collective intelligence (Riedl, 2024), marking a shift from optimizing for stand-alone AI performance to optimizing for the performance of human–AI teams.

**Limitations and Future Work.**   While we validate our framework using well-structured academic tasks (adapted from MMLU), these items represent a small subset of real-world collaborative contexts. Future work should examine whether our approach generalizes to other collaborative settings such as creative group work, workflow planning, or software development. We note that while the present study focuses on individual human–AI dyads, our framework naturally generalizes to multi-agent configurations. Additional hierarchical terms could model group-level collaborative ability in human teams or capture differential contributions from multiple AI assistants. Such settings may exhibit important emergent phenomena that are not observable in dyadic interactions. Developing and validating these multi-agent extensions represents an important direction for understanding collective intelligence in hybrid human–AI systems.

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

## Supplementary Information

## A    Background on Theory of Mind in Human-AI Settings

To explain possible sources and mechanisms of human-AI synergy we draw on established theories in human-human collaboration, in particular Theory of Mind. Theory of Mind (ToM) is the capacity to represent and reason about others' mental states—such as their beliefs, goals, or intentions—to explain and predict behavior (Premack & Woodruff, 1978). ToM plays a crucial role in human-human interaction (Nickerson, 1999; Lewis, 2003). It enables coordinated behavior by allowing individuals to anticipate others' actions and knowledge, establish common ground, disambiguate and repair communication, and adapt their contributions during joint tasks (Frith & Frith, 2006; Clark, 1996; Sebanz et al., 2006; Tomasello, 2010). ToM is the glue that binds autonomous agents into a collective epistemic agent (Clark & Chalmers, 1998; Tollefsen, 2006). It has repeatedly been shown to be a strong predictor of success in collaboration (Weidmann & Deming, 2021; Woolley et al., 2010; Riedl et al., 2021).

When teaming with generative AI like LLMs, users similarly need to anticipate their behavior and adapt accordingly given their semi-autonomous and unpredictable nature. Such adaptation is necessary as LLMs appear to users as having some level of agency and unpredictability. Three aspects of contemporary generative AI systems stand out. First, these they possess meaningful agency by generating novel content without explicit commands, creating unpredictability that requires coordination (Dong et al., 2025). Second, successful collaboration with LLM systems depends on the same dialogic mechanisms as human teamwork: eliciting outputs and providing feedback based on an evolving understanding of the partner's capabilities and goals (Clark & Brennan, 1991). Third, deciding whether to accept, reject, or further interrogate LLM responses depends on estimating its competence and "beliefs". Fourth, LLMs exhibit behavior that strongly invites attribution of intentional states, encouraging users to ascribe beliefs (Gopnik & Wellman, 1992). This tendency toward anthropomorphism is not just a cognitive bias but reflects the genuinely interactive nature of AI responses.[11]

Robust coordination with such agents benefits from users ability to form and revise mental models of the AI knowledge and capabilities and misaligned expectations about the AI's capabilities, knowledge, or "goals" is likely to hinder task performance. That is, successful AI partnerships require users to engage in dialogue with their AI partner, form accurate mental models of what the AI knows and doesn't know, and provide clear instructions informed by this understanding (cf. Laban et al., 2025, for similar arguments). When the AI makes errors, effective collaborators troubleshoot by hypothesizing what information the system lacked and updating their mental model accordingly.[12] In human-AI collaboration, ToM may enable users to form accurate mental models of the AI's capabilities and limitations, creating effective coordination (despite the AI's lack of mutual intentionality). ToM behavior may also support role differentiation, allowing users to leverage the AI's strengths, a division of cognitive labor resembling a transactive memory systems in human groups (Lewis, 2003; Wegner, 1986).

Adopting a team-based perspective on human-AI interactions yields theoretical predictions which can be evaluated with human-AI benchmarks. First, we hypothesize that performance varies across conditions in a way that reflects two distinct, condition-specific latent abilities: individual problem-solving ability when working alone and collaborative ability when working jointly with AI. Evidence from human teams suggests that these two dimensions of human ability are separately identifiable,

---

[11]The human tendency to treat nonhuman agents as humanlike and apply social heuristics and mental-state inferences to interactive systems was documented even with pre-AI software (Reeves & Nass, 1996; Nass & Moon, 2000; Weizenbaum, 1966; Waytz et al., 2010). Whether LLMs have a generalizable ToM is currently an open question, see for example (Prakash et al., 2025).

[12]This framework stands in contrast to traditional tool use, such as working with a notebook (Clark & Chalmers, 1998). A notebook functions as a pure storage device—it cannot act independently, making its "behavior" entirely predictable. Effective notebook use relies on meta-cognitive awareness of one's own memory limitations rather than theory of mind about another agent's capabilities. As a result, individuals skilled at self-monitoring and organizing information benefit most from notebooks, not those adept at understanding and coordinating with other minds. Working with a notebook, users are not required to form mental model of why the notebook inserts certain keywords, whether it will auto-correct technical terms, or when it might reorganize pages.

with each broad ability contributing independently to team success (Weidmann & Deming, 2021). Second, given the importance of ToM in enabling effective coordination and adaptive interaction in human groups, we hypothesize that higher ToM capabilities will positively predict collaborative ability, but not solo ability when working alone. Third, we hypothesize that ToM acts as a direct mechanism enabling synergy in human-AI teams. Concretely, we test the idea that users with strong ToM provide prompts that lead AI systems to engage in higher-quality dialogue. Last, we build on recent advances in psychology suggesting that ToM is a dynamic construct (Schneider et al., 2012). That is, ToM can be thought of as a stable individual trait whose expression is shaped by contextual affordances and cognitive demands. As such, we should expect ToM expression to vary within individuals (Qureshi et al., 2010; Lin et al., 2010; Meijering et al., 2010) and hypothesize that within-user dynamic *increases* in ToM expression (i.e., when users exhibit unusually strong Theory of Mind) will be positively associated with the quality of AI responses.

## B METHOD DETAILS FOR THEORY OF MIND ANALYSES

We build on the decomposition in Eq. 2 to investigate the cognitive mechanisms underlying effective collaboration between humans and AI. In principle, any user-level characteristic can be used to explore why some users have stronger collaborative abilities when working with AI. In Eq. 3, we focus on Theory of Mind. As discussed in section 1.2, evidence from literature on collective intelligence and human teams suggests that ToM is a crucial element of successful teamwork.

$$\Pr(Y_{ij} = 1) = \begin{cases} \mathrm{logit}^{-1}(\theta_i^{\mathrm{Solo}} \times ToM_i - \beta_j) & \text{if } joint_{ij} = 0 \text{ (solo)} \\ \mathrm{logit}^{-1}(\underbrace{\kappa_i^{\mathrm{Team}} \times ToM_i + \kappa_{\mathrm{AI}[i]}^{\mathrm{Team}}}_{\text{Ability}}) - \underbrace{(\beta_j + \gamma_j)}_{\text{Difficulty}}) & \text{if } joint_{ij} = 1 \text{ (with AI)} \end{cases} \quad (3)$$

We analyze each user's written prompts for signatures of Theory of Mind (ToM)—the capacity to reason about others' knowledge, beliefs, and intentions (see Appendix for description of how ToM was measured). By integrating behavioral performance with linguistic indicators of ToM, we aim to better understand how users' mentalizing abilities contribute to the emergent human-AI collaboration synergy.

Last, we explore whether user ToM impacts the quality of AI responses. First, we collect additional data on the AI response quality (see Appendix for details). We then exploit the panel data structure with repeated observations to perform a variance decomposition (Mundlak, 1978). This analysis decomposes each user's Theory of Mind signatures into a user-level mean, which captures trait-like differences across users, and a within-user deviation term, which reflects within-user, dynamic changes in ToM expression. Both elements can theoretically influence AI response quality.

We compute the user-level mean as leave-one-out measure (Jacob & Lefgren, 2008). This cross-validation style estimation of the user-level trait helps to address concerns of overfitting and avoids contamination of the latent trait estimate. Using question-level $ToM_{ij}$ measures for question $j$ by user $i$ we compute

$$\overline{\mathrm{ToM}}_{i(-j)} = \frac{1}{J_i - 1} \sum_{j' \neq j} \mathrm{ToM}_{ij'} \quad (4)$$

$$\mathrm{ToM\_dev}_{ij} = \mathrm{ToM}_{ij} - \overline{\mathrm{ToM}}_{i(-j)} \quad (5)$$

The ToM analyses include three additional control variables for effort, all computed on the question level.

**Turns.** A count of the number of back-and-forth conversation turns between the user and the AI. More turns indicate more effort, such as asking for clarification or providing refinements.

**Total Chars.** A count of the number of characters written by the user across all conversation turns of the question. Writing longer prompts (controlling for the number of prompts) indicates more effort.

**Edit Ratio.** The effort ratio as the normalized Levenshtein distance between the user's concatenated LLM prompt(s) and the original question, capturing the proportion of characters that were edited and thus reflecting the extent of user-initiated modifications. User prompts that simply copy & pasted the original question have an edit ratio of 0 and would thus indicate low effort.

## C    THEORY OF MIND MEASUREMENT

We use LLMs to perform text analysis of users' written prompts. Prior work has shown that modern LLMs can do this with high accuracy and reliability, achieving scores above the trained human coders Rathje et al. (2024). It is sometimes referred to as a Language Model as a Research Assistant (LMRA; Eloundou et al., 2024). We used the LLM prompt below and `gpt-4.1-2025-04-14` with temperature set 0. The assessment is performed on the question level, concatenating all user prompts (if there were multiple; but excluding the AI responses). The analysis builds on ideas of using computational text analysis indexing dynamic collective constructs from observed interaction text (Mathieu et al., 2022). Prior work has established the validity of analysis of recorded conversations to assess theory of mind capacity, especially in the context of collaboration success (Hennessy et al., 2016; Knight & Mercer, 2017; Kim et al., 2021; Alkire et al., 2023) and the relationship between traditional ToM measures such as performance on false-belief tests, competence at reading others' minds in everyday conversational interactions, and other ToM measures like the reading the mind in the eyes test (De Rosnay et al., 2014). The prompt design relies on chain-of-thought reasoning with in-context examples (Wei et al., 2022). The specific markers of ToM are compiled from prior work on ToM, in particular recent reviews of ToM in psychology (Beaudoin et al., 2020; Fu et al., 2023; Quesque & Rossetti, 2020), working linking ToM to rich signals in communication such as clarification requests and communication repair (Tomasello & Rakoczy, 2003), work in computer science (e.g., Bara et al., 2021), and older ideas around social intelligence (Kihlstrom & Cantor, 2000).

As a robustness test and to address concerns of style confounds or model family effects, we also collected theory of mind assessments using Google's `Geimini 2.5 Pro`—an entirely different family of model, trained by a different research lab—and find substantially the same results (see footnote 7).

---

**ToM Signatures in User Prompts**

Context: You are an expert research assistant.
Evaluation Task: You are evaluating the degree to which the dialogue by a human user with an AI assistant show signs of theory of mind (ToM) of the human user. Theory of Mind refers to the ability to attribute mental states—such as beliefs, intents, desires, or knowledge—to oneself and others. Evaluate questions on a scale from 0 (no sign of theory of mind) to 5 (multiple signs of theory of mind present). In addition to the question asked by the user (USER DIALOGUE), you are also provided the original question (ORIGINAL QUESTION) that the user is trying to answer with the help of AI so that you can assess if and how the user has modified their query. In your evaluation, consider the following signs of theory of mind.
Indicators of high theory of mind:

- Establish rapport: e.g., by introducing themselves ("hello"), by acknowledging responses (such as "thanks" and "okay").

- Belief tracking: e.g., inference of AI's beliefs about world state and tracking how they may change dynamically during conversation.

- Knowledge gap detection: Identifying information asymmetries between the user and the AI.

- Perspective-taking: Understanding AI's visual and spatial reasoning, and that it may differ from the user's own. Awareness about what the AI needs to know (vs. what is irrelevant) to work on the problem together.

- Goal inference: Predicting the AI assistant's intended goals from dialogue.

- Communication intent: Reasoning about why specific information was shared or what the users goals are (e.g., "Help me solve this questions", "I already solved the problem").

- Mental state attribution: Understanding that the AI as unique knowledge, capabilities, and limitations.

- Plan coordination markers: Linguistic signals of strategy alignment and developing a collaboration plan (e.g., "I'm going to ask you a question").

- Explanatory dialogue: Information provision addressing knowledge gaps such as providing context about their own knowledge level (e.g., "I'm a beginner in physics"), setting expectations (e.g., "I need a comprehensive explanation"), and goals (e.g., "Explain this like I'm 15").

- Confirmation seeking: Verification behaviors indicating uncertainty, including asking for explanations and clarification (e.g., "Is my approach correct?").

- Positioning and coordination: Awareness and coordination of different viewpoints

- Build on ideas: Understanding of others' mental states with elaboration (e.g., "this is not what I meant" and "the answer should be formatted as a fraction").

- Reference back: Explicit acknowledgment showing memory of partners' thoughts

- Challenge/disagree: Sophisticated engagement demonstrating ToM understanding such as pointing out mistakes (e.g., "could you be wrong?").

- Epistemic markers: Language revealing understanding of knowledge states ("I think you know...")

- Justification requests: Questions showing awareness of reasoning processes

- Evaluative language: Understanding of partners' credibility assessments

- Metacognitive references: Explicit discussion of thinking processes (e.g., "right, that is a difficult problem").

- Conversational adaptation: Real-time adjustment to partner's communication style.

Absence of the above "high theory of mind" markers should be classified as low levels of theory of mind. Additionally, consider these indicators of low theory of mind:

- Sharing irrelevant information: failing to detect relevant knowledge asymmetries.

- Assuming shared context without previously establishing it (e.g., saying "answer this question" without specifying what the question actually is).

- Delegating a trivial question (e.g., "how many years are in a decade") or moral questions for which the human user clearly has the advantage (e.g., "is peeing in a pool bad") are signs of low theory of mind.

- Misunderstanding the AI's unique knowledge, capabilities, and limitations such as treating it as Google (e.g., "stomach hurts sharp pain" indicates the user is search for a website rather than asking a specific question that is more aligned with the AI's capabilities).

Instructions:
1. Analyze whether the question shows signs of theory of mind thinking.
2. Clearly explain your reasoning in a paragraph.
3. Then, in a separate line, output the classification in the following format:
FINAL ANSWER: [0-5]

## D   VALIDATION USING HUMAN RATINGS

To validate the approach of studying dialogue-level ToM signatures in human–AI collaboration and the LMRA assessment, we collected gold standard human ratings for a stratified (by ToM level) random sample of 120 dialogues. We recruited 32 subjects from Prolific (general US pool), each of them rated 19 dialogues on average (SD=14) yielding an average of five ratings per dialogue. We paid human raters $1.50 for reading the instructions, $0.25 per rated dialogue, and a $1.50 bonus if they rated more than 10 dialogues (to encourage raters to evaluate several dialogues to increase consistency). The rating instructions were mostly identical to those uses for the LLM with minor changes to formatting (e.g., breaking instructions up into several pages) and the addition of information about compensation.

Individual human raters showed low reliability (ICC2 = 0.13), reflecting some inherent noise in single ratings. The aggregated gold standard showed excellent reliability (ICC2k = 0.83 [95% CI: 0.78 - 0.87]). The high inter-rater reliability among independent human judges demonstrates that dialogue-level ToM signatures in human–AI collaboration can be assessed consistently, validating the construct and supporting the use of this evaluation approach. The LMRA's ratings demonstrate substantial agreement with the gold standard (ICC2 = 0.42; ICC2k = 0.59 [0.42 - 0.72]), outperforming a single human rater and approaching the consistency of the aggregated human consensus. All ICC estimates between the LMRA and the human gold standard are significant ($p < 0.001$). These results support the use of a LMRA as a proxy for human assessment, and demonstrate the promise of LLM-based assessments as scalable approximations of human judgment for dialogue-level ToM. However, reliability of individual human ratings is low and the findings may be task-specific.

## E  CONSTRUCT VALIDITY OF LMRA TOM RATINGS

To further establish the construct validity of LMRA ToM ratings of user dialogues we took several additional steps to implement emerging LLM-as-a-judge best practices (Li et al., 2025). First, we inspected dialogues for signs of adversarial prompt injection (Maloyan & Namiot, 2025) but found no signs of it. This is not surprising given that users were unaware that we would code their dialogues for signs of ToM and that they were incentivized to achieve high question answering performance. Next, to address concerns of style confounds or model family effects, we replicated data collection using four other models: Google's `Gemini 2.5 Pro`, Anthropic's `Claude Opus 4`, OpenAI's `gpt-oss-120b`, Alibaba's `Qwen3 32B`, and DeepSeeks `Deepseek R1`. To assess robustness to prompt bias, we also collected data using an expanded ToM-assessment instruction set and alternative rating schemes (point-based assessment vs. pairwise comparisons). To explore ordering bias, we collected data using a prompt which reverses the instructions for high vs. low ToM signatures. Across all nine LMRA "raters" we find excellent (Cicchetti, 1994) ensemble inter-coder reliability (ICC(2,k) = 0.90 [0.85 - 0.92]). We also show all pairwise ICC2 scores (Figure A1). The smaller `Llama-3.1-8b` shows substantially lower agreement with the other models, consistent with limited capacity to judge ToM reliably. As a sensitivity analysis, we exclude this model which increases ICC(2,k) to 0.92 [0.91 - 0.94]. Taken together, these findings provide convergent evidence that LMRA ToM ratings validly capture the intended construct, are robust across models and prompts, and suggest that any single model's ratings are representative of the ToM construct.

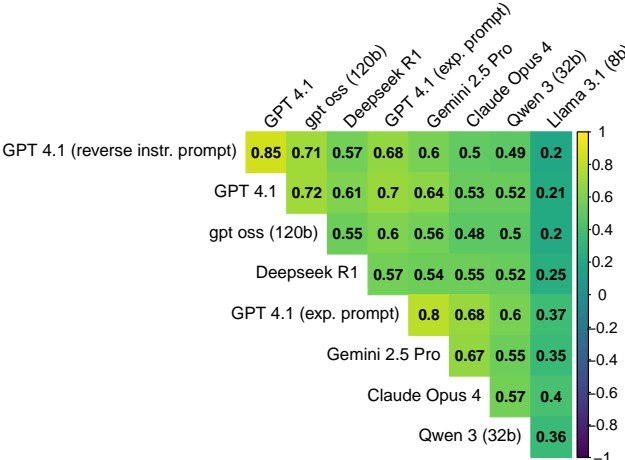

Figure A1: **Pairwise inter-coder reliability (ICC2) of LMRA ToM assessment.**

Finally, to test robustness to judgment bias we collected data using a prompt that asked the LMRA to make pairwise comparisons as to which dialogue shows higher degree of ToM (instead of asking for assessment on 0-5 rating scale). We collected data on 12,000 pairwise comparisons drawing two random dialogues using `gpt-oss-120b`. Next, we consider these pairwise judgments "matches" between the users who wrote them and aggregate them in an Elo-based ranking (ranking all users in the study by their ToM prompt writing capacity; based on about 37 pairwise judgments per user).

We find moderate-to-strong rank association between the baseline `GPT 4.1` average rating and the Elo-based ranking ($\rho_s = 0.61$; $p = 2.2 \times 10^{-16}$). After calibrating the Elo scores to the 0–5 rating scale (via linear regression), we find moderate-to-good absolute agreement ICC(3,1) = 0.68 [0.64 - 0.72] with `GPT 4.1` scores and ICC(3,1) = 0.77 [0.73 - 0.80] with `gpt-oss-120b`. These results indicate that the ToM construct can be reliably assessed using both absolute vs. relative judgment prompts.

## F    AI RESPONSE QUALITY MEASUREMENT

To assess the quality of AI responses we used `gpt-4.1-2025-04-14` with temperature set 0 and the prompt prompt below. The assessment was done on the question level, concatenating all LLM responses if there were multiple turns of interaction (but excluding the user's prompts).

---

**AI Response Quality**

Context: You are an expert research assistant. A user is working with an AI assistant to answer a question (see ORIGINAL QUESTION below). The AI gave the below response (AI RESPONSE).
Evaluation Task: You are evaluating the quality of the AI response given to the user. Evaluate AI responses on a scale from 0 (low quality, not helpful to answer the question) to 5 (high quality, very helpful to answer the question). Consider not only whether the answer itself has the correct answer but how helpful the answer is to the user to correctly answer the question (e.g., by providing explanation and reasoning).
Instructions:
1. Analyze the quality of the AI response.
2. Clearly explain your reasoning in a paragraph.
3. Then, in a separate line, output the quality assessment in the following format:
FINAL ANSWER: [0-5]
ORIGINAL QUESTION: {...}.
AI RESPONSE: {...}

---

## G    CAUSAL EFFECT OF WORKING WITH AI

The experimental design places users in the solo condition first and the with-AI condition second. One concern resulting from this design is that this may confound working with AI with temporal learning or fatigue effects since working whit AI is always observed second. As a robustness test we leverage another feature of the Chang et al. (2025) dataset. Specifically, the design included another randomly assigned treatment condition in which some users worked on a set of questions alone for a second time, before finally working with AI in the last step. Instead of modeling only two phases (solo vs. joint) we model three phases (solo vs. solo after practicing vs. joint with AI). This model explicitly control for and models any learning/fatigue effects. It thus avoids the confound between the AI condition and "AI is always observed second". We find consistent results with a slight negative effect of the second phase indicating fatiguing (rather than learning). This indicates that our main effects somewhat underestimated the true benefit of working with AI ($\rho_p = 0.58$, $p < 0.001$). This model causally identifies AI effects (from random assignment of users to AI models), phase-based learning/fatigue effects (based on phase progression with random assignment to conditions).

We also perform additional robustness tests to confirm stable estimates of individual ability using different diagnostic tests. First, MCMC diagnostics show excellent convergence (Rhat = 1.00) and very high effective sample sizes (Tail_ESS between 7,000 - 26,000 effective samples per parameter), confirming that posterior estimates of user-level ability and phase-specific slopes are stable and reliable, even though each user completed only a limited set of items per phase. Furthermore, posterior estimates correlate strongly with raw accuracy ($r = 0.75$), demonstrating that the hierarchical model produces precise, interpretable ability estimates.

## H  ADDITIONAL INSIGHTS INTO MECHANISMS OF COMPLEMENTARITY

To provide an additional window into the mechanism of complementarity we use model-based, item-level decomposition to explore item-level differences across difficulty and task domain. Specifically, we quantify shifts from answering incorrectly alone to answering correctly with the help of AI via posterior predictive counterfactuals from the fitted hierarchical IRT model (Eq. 2). This has the added advantage that we can compute counterfactuals for each user-question for both models (GPT-4o and Llama-3.1-8B).

On average, users switch from wrong answers to right answers on 25% of the tasks when working with Llama-3.1-8Band 29% when working with GPT-4o, indicating a substantial share of cases where collaboration moves responses from incorrect to correct. Switch rates increase monotonically with item difficulty for both models. On easy questions (1st decile) users switch 17% (Llama-3.1-8B) and 19% (GPT-4o). On hard questions (10th decile) users switch 31% (Llama-3.1-8B) and 38% (GPT-4o). This pattern is precisely what complementarity predicts: collaboration is most beneficial on harder items and in line with our results shown in Figure 2a.

Breaking out complementarity by domain we find the biggest complementarity for physics questions (32% (GPT-4o) and 28% (Llama-3.1-8B)), followed by math (29% (GPT-4o) and 25% (Llama-3.1-8B)), and moral reasoning (26% and 22%). The Llama-3.1-8Bvs. GPT-4o difference is consistently about 4% across the topic domains (this is in line with our model-based analyses where we do not find a significant interaction effect between models and item difficulty beyond the magnitude difference of GPT-4o being the better model overall).

