# OpenReview forum: "Quantifying Human-AI Synergy"
_ICLR.cc/2026/Conference — Submitted to ICLR 2026_

### Official Review · Reviewer_e8HW · 2025-10-31

**Soundness:** 2
**Presentation:** 2
**Contribution:** 1
**Rating:** 2
**Confidence:** 4

**Summary:**

This paper proposes a Bayesian Item Response Theory (IRT) framework to quantify human–AI synergy, separating individual from collaborative ability while controlling for task difficulty. The authors demonstrate the framework on two datasets and AI models. The results show that AI helps improve human-AI accuracy. The paper also uses Theory of Mind (ToM) to interpret the observed human-AI synergy.

**Strengths:**

- The authors present the results grounded in empirical results and connect well to prior benchmarks.
- The analysis using Theory of Mind provides an interesting bridge between computational modeling and social cognition.

**Weaknesses:**

- The model structure is oversimplified compared to the ambitious contribution claimed by the authors. It only considers the ability and the  difficulty. A lot of factors are ignored such as learning effects. The assumption on additivity of ability is also unrealistic--there are a lot of case where human and AI are substitute or complementary with each other.
- The choice of the Bayesian model is not clear. Since the Bayesian workflow is often iterative and involves model fitting and then checking, the authors should do more model comparison to motivate the model specification they arrive at.
- The experiments also simplify a lot than realistic human-AI collaboration scenarios. For example, three questions done alone with AI is not much to get a good estimate of ability in my opinion.
- The paper does not engage with prior work on human–AI complementarity enough. For example, how the framework improves interpretability over existing regression-based or causal models of AI assistance effects?

**Questions:**

- What are the assumptions of the framework? In general, I would suggest the authors to lay out the assumptions that must hold for their results to be easy to interpret.

---

> ### Author Response · Authors · 2025-11-16
> **Re: Oversimplified model structure => random effects capture systematic heterogeneity in user ability and item difficulty**
>
> We appreciate the reviewer’s concern about model simplicity. Our primary inferential strategy deliberately models both user-level and task-level variation: the model includes both user-level and task-level random effects (random intercepts & random slopes), which capture systematic heterogeneity in user ability and item difficulty and absorb any unmeasured person- and item-specific confounders (e.g., cognitive ability, prior experience with the task, domain knowledge, or item clarity or ambiguity). Learning effects are explicitly represented by the experimental three-phase design and by allowing user-phase-specific random slopes. That is, the model directly estimates how performance changes across phases at the individual level rather than assuming a single average learning curve. The fitted model shows large user-level variance in baseline accuracy (SD = 1.25) and in learning slopes for phase 2 (SD = 0.79) and phase 3 (SD = 1.22). Importantly, these slopes are strongly correlated with baseline ability (e.g., Corr(Intercept, phase 3) = –0.83), indicating that individuals differ systematically in how much they learn.
>
> The model explains about 34% of the variance in the latent propensity to answer correctly (logit scale). This is substantial for a logistic IRT model and indicates that the hierarchical structure captures a large share of systematic variation.
>
> These findings confirm that both users and tasks respond very differently across phases, and the hierarchical model captures this non-additive, non-uniform structure. Far from being a “simple” model, the random-effects specification explicitly models the complex heterogeneity the reviewer highlights.

---

> ### Author Response · Authors · 2025-11-16
> **Re: choice of the Bayesian model is not clear => Bayesian workflow and model comparison**
>
> We thank the reviewer for raising this point. Our modeling strategy follows the standard Bayesian workflow described in Gelman et al. (2020): we begin from the data-generating structure implied by the experiment (binary responses nested within users and tasks), specify a hierarchical logistic model consistent with IRT practice, check priors, evaluate posterior predictive fit, and compare against simpler alternatives. In particular, we explicitly compared a random-intercepts-only model with a random-intercepts-plus-slopes model and found that the latter provided substantially better predictive performance (LOO difference = 50.9, SE = 10.2). Bayesian workflow does not require exhaustive combinatorial model search; instead, it emphasizes prior checks, model checking, and considering alternative formulations when diagnostics suggest misfit. The hierarchical Bayesian models we employ also already encode model regularization and partial pooling that make classical model-comparison-driven variable selection far less necessary.
>
> Our posterior predictive checks, variance decompositions (ToM as latent factor vs. time-varying ToM ability), and model comparisons (base model from Eq (1), model with AI fixed effects Eq (2), model with phases to capture learning (Appendix Section G), model with ToM moderation (Eq 3), single latent ability vs. separate abilities) all indicate that the reported models are appropriate for the design and supported by the data.
>
> Gelman, A., Vehtari, A., Simpson, D., Margossian, C. C., Carpenter, B., Yao, Y., ... & Modrák, M. (2020). Bayesian workflow. arXiv preprint arXiv:2011.01808.

---

> ### Author Response · Authors · 2025-11-16
> **Re: Only three items per user as basis for ability => robustness tests confirm stable estimates of individual ability**
>
> We recognize the reviewer’s concern that only three items may be limited to precisely estimate user-level ability. However, our hierarchical model uses partial pooling to borrow strength across users and items, and we empirically verified the stability of person-level estimates. We find that user abilities show substantial heterogeneity (SD ≈ 0.85) with median posterior SE ≈ 0.91, demonstrating that the hierarchical model produces stable, interpretable estimates even with few items per user. A ratio near 1 indicates a reliable signal: estimates are meaningful but appropriately shrunk – given our setting with few items per users this is good. MCMC diagnostics show excellent convergence (Rhat = 1.00) and very high effective sample sizes (Tail_ESS 7k-26k effective samples per parameter), confirming that posterior estimates of user-level ability and phase-specific slopes are stable and reliable, even though each user completed only a few items per phase. Furthermore, posterior estimates correlate strongly with raw accuracy (r = 0.75), demonstrating that the hierarchical model produces precise, interpretable ability estimates.

---

> ### Author Response · Authors · 2025-11-16
> **Re: Engage with literature regarding human-AI complementarity => refinements added to Related Work**
>
> We appreciate the reviewer’s comment regarding human-AI complementarity. Our paper explicitly builds on this literature by modeling user- and item-level heterogeneity in the effects of AI assistance, which is central to understanding complementarity (e.g., when AI helps low- vs. high-ability individuals more); which tasks benefit disproportionately from AI assistance; and when learning interacts with assistance. We have revised the related-work section to connect our modeling approach more explicitly to prior work on complementarity in human-AI teams, including studies on skill substitution/complementarity (e.g., Malone et al.; Dell’Acqua et al.; Wilder et al.), adaptive assistance, and differential susceptibility to AI support.
>
> Our hierarchical Bayesian model improves interpretability relative to standard regression or causal models in several ways: (1) It decomposes performance into individual ability, task difficulty, and experimentally manipulated assistance effects (different AI models), enabling a clean attribution of where complementarity arises; (2) The model quantifies individual-level heterogeneity in assistance effects (random slopes), directly identifying who complements AI and who substitutes it; (3) Partial pooling prevents overfitting and yields calibrated estimates for each user-AI interaction; and (4) phase-specific effects allow us to isolate learning-based complementarity. We have revised the related work section to more directly engage with the existing literature on human-AI complementarity as you suggested.

---

> ### Author Response · Authors · 2025-11-16
> **Re: What are the assumptions of the framework? => done: new content added**
>
> This is very helpful feedback. One of our core goals is to give people a framework that yields interpretable results. In response we’ve added a section to the paper that clarifies the assumptions required for interpretability. We write the following:
>
> Our analytical framework relies on several assumptions and data requirements. First, the Item Response Theory structure assumes monotonicity and unidimensionality. We assume respondents have a single latent ability when they’re working alone, or working with AI --- and that the probability of correct answers monotonically increases with ability.
>
> Second, to identify and estimate solo and collaborative ability our setup requires item crossing: i.e. that the same questions are tackled by participants working alone, and working with AI. The model then assumes that the decomposition of difficulty is additive, i.e., that collaborative difficulty supplements (but does not qualitatively alter) the underlying difficulty of each item.
>
> Third, our framework tests—rather than assumes–whether joint human–AI performance can be decomposed into additive contributions of the human and the AI model. We treat human collaborative ability and model-specific collaborative capability as separable components of team performance. This implies that interactions between particular users and particular models do not introduce systematic idiosyncratic effects beyond those captured by random variation.
>
> Finally, the Bayesian hierarchical structure assumes appropriate partial pooling across users and items, enabling stable estimation under sparse or uneven data. This approach presumes that user- and item-level effects are exchangeable within their respective distributions.
>
> Together, these assumptions allow us to interpret our estimates of solo ability, collaborative ability, task difficulty, and AI-induced performance gains as reflecting stable and separable components of human–AI collaboration.

---

> ### Author Response · Authors · 2025-11-27
>
> We would like to thank the reviewer again for their feedback. We wanted to check whether our response addresses your concerns. We're happy to provide further clarification if needed.

---

### Official Review · Reviewer_bPAp · 2025-11-01

**Soundness:** 3
**Presentation:** 4
**Contribution:** 3
**Rating:** 8
**Confidence:** 4

**Summary:**

In this paper, the authors propose a Bayesian Item Response Theory (IRT) framework for evaluating human–AI collaboration. Instead of measuring model performance in isolation, the proposed approach jointly models human and AI contributions during cooperative task solving. The authors apply this method to the ChatBench dataset, where 667 participants complete multiple-choice tasks under both solo and AI-assisted conditions. The results indicate that collaborative performance can differ from solo capability and varies across individuals. The paper also examines Theory-of-Mind (ToM) scores and reports that higher ToM ability is associated with greater improvement when using AI assistance.

**Strengths:**

1. This paper tackles a timely and important problem, as understanding interactive LLM behavior and human–AI teaming is increasingly critical when transitioning from offline evaluation to real-world deployment scenarios.

2. The work provides a strong methodological contribution by proposing a principled Bayesian IRT framework that decomposes human solo ability, human–AI collaborative ability, AI contribution, and task difficulty to quantitatively measure human–AI synergy, and the method is supported by strong empirical evaluations.

3. The study offers interesting insight into cognitive mechanisms underlying human–AI interaction, as the finding that Theory-of-Mind predicts collaborative gain, not solo performance, helps explain when and why human–AI synergy emerges, which also provides valuable implications for practitioners on how AI systems should be designed to better support human decision-making.

**Weaknesses:**

1. Experimental tasks all fall within academic contexts  (MMLU-adapted questions). Although justified as a structured benchmark, future work could validate synergy in more naturalistic settings (e.g., group creative work, travel planning, or collaborative coding).

2.  ToM scoring uses an LLM rater, which is reasonable given recent literature on LLM-as-judge, but still vulnerable to construct validity concerns. The authors partially address this via human validation, but a deeper discussion (e.g., potential LLM bias or adversarial prompt cases) in cases where such human and llm alignment diverges significnantly would strengthen this part.

3.  Limited to individual human–AI teaming. It would be interesting to see whether the method generalizes to multi-agent settings where a single human interacts with multiple AI assistants or where a group of humans collaborates with one AI system.

**Questions:**

See above weakness.

---

> ### Author Response · Authors · 2025-11-15
> **re 1. Experimental tasks all fall within academic contexts -- we fully agree => discussion expanded**
>
> Excellent point! We fully agree that our empirical example focuses on well-structured academic tasks and does not capture the full breadth of real-world collaborative demands. The goal of the paper is to isolate and validate the analytical framework under controlled conditions where task difficulty and user ability can be rigorously estimated. We agree that an important next step is to examine whether synergy manifests in naturalistic, real-world settings. We have added a discussion of this limitation and outline how the framework can be extended to richer, less structured forms of collaboration.
>
> We have also added new analyses in the Appendix breaking out complementarity by task domain. We find the biggest complementarity for physics questions (32% (GPT4o) and 28% (Llama)), followed by math (29% (GPT) and 25% (Llama)), and moral reasoning (26% and 22%).
>
> == Added to Limitations ==
> While we validate our framework using well-structured academic tasks (adapted from MMLU), these items represent a small subset of real-world collaborative contexts. Their structured nature provides experimental control and enables precise estimation of user ability and item difficulty, but limits ecological validity. Future work should examine whether our approach generalizes to more naturalistic collaborative settings such as creative group work, workflow planning, collaborative writing, or software development. Extending the framework to open-ended tasks would also test whether the additive ability decomposition we propose holds when interactions become more fluid, strategic, or socially complex.

---

> ### Author Response · Authors · 2025-11-15
> **re 2. ToM scoring uses an LLM rater -- clarification and acknowledgement in limitations**
>
> This is helpful feedback, thanks. We agree that while LLM-based coding is increasingly reliable, construct validity can’t be taken for granted. We have significantly expanded our analyses to address construct validity concerns. First, we collected data from 5 additional models (Gemini 2.5 Pro, Claude Opus 4, gpt-oss-120b, Qwen3 32b, and DeepSeek R1). To assess robustness to prompt bias, we also collected data using an expanded ToM‑assessment instruction set and alternative rating schemes (point-based assessment vs. pairwise comparisons). To address ordering bias, we collected data from a prompt in which instructiosn about what constitutes high vs. low ToM were reversed. Across all nine LMRA ``raters'' we find excellent ensemble inter-coder reliability (ICC(2,k) = 0.90 [0.85 - 0.92]). Individual pairwise rating consistency is shown in a new heatmap figure. Finally, we test robustness to using LLM judges to make absolute vs. relative judgment. Collecting 8,000 pairwise judgments “which dialogue shows higher ToM”, we aggregate these pairwise judgments into Elo-based scores. We find moderate-to-good absolute agreement ICC(3,1) = 0.63 [0.58 - 0.67] with GPT 4.1 scores and ICC(3,1) = 0.71 [0.67 - 0.74] with gpt-oss-120b. These results indicate that the ToM construct can be reliably assessed using both absolute vs. relative judgment prompts, which are robust across different models and other prompt changes. These new analyses are in a new section in the Appendix titled Construct Validity of LMRA ToM Ratings.
>
> We have expanded our discussion to highlight potential LLM biases, sources of construct drift, and scenarios in which human–LLM alignment may break down (see discussion, below). We also agree that future work should investigate robustness under adversarial prompting, linguistic style variation, and cross-cultural conversational norms.
>
> == added to Limitations ==
> While LLM-based evaluation offers scale, consistency, and cost efficiency, it is not immune to construct validity concerns. LLM raters may exhibit systematic biases and be vulnerable to adversarial prompt structures that mimic ToM markers without reflecting genuine mentalizing. Although our human-rating validation shows strong alignment at the aggregate level, divergence in more challenging or atypical cases remains possible. Future work should systematically probe these edge cases and evaluate robustness across model families, prompting strategies, and conversational styles.

---

> ### Author Response · Authors · 2025-11-15
> **re 3. Limited to individual human–AI teaming -- we couldn't agree more! ==> our framework would work for that too**
>
> We couldn’t agree more with this excellent point. Our present focus on individual human–AI dyads reflects both the structure of ChatBench and the need to introduce and validate the framework in its simplest identifiable form. But indeed, insisting on a single-user baseline restricts such analyses to specific types of tasks and leaves broader questions about human-AI collaboration in larger collectives unanswered. We agree that extending the model to multi-agent settings is a natural and exciting direction. The additive and hierarchical structure of our model is compatible with such extensions: additional random effects or interaction terms could capture group-level collaborative ability, transactive memory structures, or division of labor across multiple agents. We now explicitly discuss how the framework could generalize to multi-agent or multi-assistant systems in the discussion. Thanks for this prompt!
>
> == added to Discussion ==
> Last, we note that while the present study focuses on individual human–AI dyads, our framework naturally generalizes to multi-agent configurations. Additional hierarchical terms could model group-level collaborative ability in human teams or capture differential contributions from multiple AI assistants. Such settings may exhibit important emergent phenomena that are not observable in dyadic interactions. Developing and validating these multi-agent extensions represents an important direction for understanding collective intelligence in hybrid human–AI systems.

---

> ### Author Response · Authors · 2025-11-27
>
> We would like to thank the reviewer again for their feedback. We wanted to check whether our response addresses your concerns. We're happy to provide further clarification if needed.

---

### Official Review · Reviewer_1WjX · 2025-11-01

**Soundness:** 2
**Presentation:** 4
**Contribution:** 3
**Rating:** 4
**Confidence:** 4

**Summary:**

The paper proposes a method to measure human–AI synergy, where synergy is defined as how much an AI partner improves human performance. The framework follows Bayesian Item Response Theory (IRT). It separates solo human ability, collaborative ability with AI, and item difficulties in solo vs. joint settings. Using ChatBench (396 MCQs across math, physics, moral reasoning; 667 participants), the study shows that human–AI teams outperform humans alone or AI alone, and benchmarks models by their average improves human performance while controlling for difficulty and user abilities. Finally, the authors test whether Theory of Mind (ToM) in users explains who benefits most and find ToM predicts collaborative performance.

**Strengths:**

* Clear synergy metric for model benchmarking. The approach directly estimates each model’s capacity to raise the performance of the average user, rather than relying on static model-alone accuracy, enabling apples-to-apples comparisons of “collaborative capability.”


* ToM as a cognitive mechanism for teaming. Users with higher ToM do better with AI but not alone; the paper frames ToM as a plausible mechanism for coordination and division of cognitive labor, aligning with established social-cognitive theory and giving a concrete lens for why teaming helps.

* Empirical improvement that is practically meaningful. The paper finds human-AI even beats AI alone in the descriptive analysis.

**Weaknesses:**

* Framework builds heavily on prior modeling choices. The novelty is mainly in applying existing method to human–AI collaboration.

* ToM effects appear small-to-moderate. My main concern with the findings in the paper is that the ToM–collaboration link is statistically positive but not large (e.g., Spearman ~ 0.17 for joint ability, significant; ~ 0.06 and n.s. for solo). The results seem to suggest ToM is useful, but one factor among several.


* Limited window into mechanisms of complementarity. While the paper argues that ToM enables coordination, it does not decompose how human and model contributions combine at the item level (e.g., how many tasks are correct in human-AI collaboration but wrong in both alone?).

**Questions:**

* Quantifying complementarity directly. Since human+AI outperforms each constituent, what fraction of items are: wrong for both solo agents but right as a team or right for one party but not the other? Reporting these rates would put concrete numbers on complementarity.

* Break out complementarity by difficulty deciles and domain. I would guess one reason for complementarity is human provide reasoning steps to AI. It would be helpful to clarify if AI baseline uses reasoning.

---

> ### Author Response · Authors · 2025-11-17
> **re: ToM effects appear small-to-moderate => when put in comparison to human-human studies they are strong**
>
> That is a fair point – a 0.17 is not huge, but it is substantively meaningful. To see this, let us put this in perspective: the traditional way of measuring ToM (via the Reading the Mind in the Eyes test) is one of the best known predictors of collaboration performance in human-human teams (Riedl et al., 2021; Weidmann & Deming, 2020). In a large meta-study using data from 5,279 individuals in 1,356 teams, Riedl et al. (2021) find that ToM predicts about 3% variation in team performance. A ~0.17 Spearman correlation (or equivalently about 0.18 Pearson correlation) corresponds roughly to 3.3% explained variation. So our 3.3% explained variation is about exactly what we would expect from prior research on the ToM-collaboration link. Furthermore, Riedl et al. (2021) provide additional context for this number (Fig. 3 in their paper): our predictive power of 3.3% is roughly on par with the predictive power of team size. Only individual skill and team processes are better predictors of variation in team performance. That is to say: yes, 0.17 seems low, but explaining variation in team performance is extremely challenging and so ~3% is comparable to known state-of-the-art predictors.

---

> ### Author Response · Authors · 2025-11-17
> **re: Limited window into mechanisms of complementarity => done - expanded analysis**
>
> Thank you for pushing us to make the complementarity mechanisms more concrete and focus on item-level complementarity. We have added a model-based item-level decomposition that directly targets your questions. We cannot report realized overlaps for the same person-item across both with-AI and alone conditions because those counterfactuals do not exist: users answered a given question either with the AI or alone. Instead, we quantify these overlaps via posterior predictive counterfactuals from the fitted hierarchical IRT model. This has the added advantage that we can compute counterfactuals for each user-question for both models (GPT 4o and Llama). Based on these counterfactual estimates, we can now report complementarity by difficulty deciles and domain in the way you ask for.
>
> On average, users switch from wrong answers to right answers on 25% of the tasks when working with Llama and 29% when working with GPT 4o, indicating a substantial share of cases where collaboration moves responses from incorrect to correct. Switch rates increase monotonically with item difficulty for both models. On easy questions (1st decile) users switch 17% (Llama) and 19% (GPT 4o). On hard questions (10th decile) users switch 31% (Llama) and 38% (GPT 4o). This pattern is precisely what complementarity predicts: collaboration is most beneficial on harder items and in line with our results shown in Figure 2a. These raw numbers are very instructive, and suggest complementarity is biggest on the hardest items. We have included these descriptives in the results. Thank you for the suggestion!
>
> Breaking out complementarity by domain we find the biggest complementarity for physics questions (32% (GPT4o) and 28% (Llama)), followed by math (29% (GPT) and 25% (Llama)), and moral reasoning (26% and 22%). The Llama vs. GPT 4o difference is consistently about 4% across the topic domains (this is in line with our model-based analyses where we do not find a significant interaction effect between models and item difficulty beyond the magnitude difference of GPT 4o being the better model overall).

---

> ### Author Response · Authors · 2025-11-27
>
> We would like to thank the reviewer again for their feedback. We wanted to check whether our response addresses your concerns. We're happy to provide further clarification if needed.

---

### Author Response · Authors · 2025-11-27

We thank all reviewers for their thoughtful and constructive feedback. We have incorporated your suggestions into the revised manuscript and uploaded an updated version. Your recommendations have significantly strengthened the work. We summarize our clarifications and updates below:

 - **Expanded analysis of complementarity by difficulty deciles and domain:** We have added a model-based item-level decomposition that allows us to break out complementarity by difficulty and domain. For example, we find the biggest complementarity---switching from wrong answers to right answers---for physics questions (32% (GPT4o) and 28% (Llama)). The new results are in Appendix H.

 - **Expanded validation of LLM-as-judge ratings of ToM:** We significantly expanded our analyses of construct validity for the assessment of ToM in user dialogues using (a) additional LLMs (Gemini 2.5 Pro, Claude Opus 4, gpt-oss-120b, Qwen3 32B, and Deepseek R1), (b) alternative prompts to address concerns of order bias, and (c) absolute vs. relative assessment using an Elo-based assessment. Overall, these findings provide convergent evidence that LLM ToM ratings validly capture the intended construct, are robust across models and prompts.

- **Expanded robustness tests and discussion of modeling choices and assumptions:** We clarified and validated the hierarchical Bayesian modeling approach, added robustness tests demonstrating robustness of individual-ability estimates despite few items, documented the Bayesian workflow with model comparisons and posterior checks, expanded engagement with the human-AI complementarity literature, and added a new section detailing assumptions for interpretability.

We appreciate the reviewers’ insights and believe that these revisions significantly strengthened our work to better understand real-world human-AI complementarity and the role of ToM in predicting these collaborative gains.

---

### Meta-Review · Area_Chair_t4Jv · 2026-01-11

**Summary:**

Despite some positive feedback regarding the presentation and the timeliness of the topic, there are some significant methodological concerns raised by Reviewer e8HW and Reviewer 1WjX.

1. Oversimplification of the Model: Though Reviewer bPAp sees it as a strength, Reviewer e8HW argued that the proposed Bayesian IRT framework is oversimplified for the ambitious claims made. Specifically, the assumption of additivity (that human and AI abilities sum linearly) is viewed as unrealistic given that human-AI interaction often involves complex substitution or complementarity dynamics that a simple additive model may fail to capture.

2. Robustness of Estimation: There are strong concerns regarding the reliability or rubustness of estimating "solo human ability" based on only the questions (Reviewer e8HW). While the authors rely on partial pooling to stabilize estimates, the critic argues this is insufficient data to form a ground truth baseline for complex ability, undermining the subsequent synergy calculations.

3. Effect Sizes and Practical Significance: Reviewer 1WjX noted that the Theory of Mind (ToM) effects, while statistically significant, are small (Spearman ~0.17). This raises questions about the practical utility of ToM as a primary explanatory mechanism for synergy in this context.

4. Ecological Validity: Both Reviewer bPAp and Reviewer e8HW noted that the reliance on academic, multiple-choice tasks (MMLU) limits the generalizability of the findings to realistic, open-ended human-AI collaboration scenarios.

**Reviewer Concerns:**

The authors provided a detailed rebuttal, but it is not not clear if they would resolve the core methodological disagreements.

The authors attempted to address concerns about using LLMs as raters by conducting robustness checks with five additional models and different prompting strategies, showing high inter-coder reliability. The authors provided new data decomposing item-level performance, showing how switch rates (from wrong to right) increase with task difficulty. This partially addressed the request for more concrete evidence of complementarity. The authors added a section explicitly listing the assumptions of their framework (monotonicity, unidimensionality, additivity) as requested.

While the authors defended their model using random effects to capture heterogeneity, the fundamental disagreement regarding the validity of the additive assumption remains. The reviewer views this as a conceptual flaw that statistical convergence diagnostics cannot explain away. The authors argued that Bayesian shrinkage makes the 3-item estimates stable. However, the concern is that stable estimates are not necessarily accurate representations of a user's true latent ability, and this foundational weakness persists. Finally, the critique that the paper is essentially applying existing IRT methods to a new domain without significant methodological innovation remains concerning.

**Reviewer Scores:**

Reviewer 1WjX  likely would have maintained their score. They would appreciate the new data on complementarity but might remain negative regarding the effect sizes and the novelty of the method.

Reviewer bPAp, the positive reviewer, might have lowered their score a bit if they had participated in a discussion regarding the "additivity assumption" raised by Reviewer e8HW.

Reviewer e8HW would likely maintain their score of 2. Their critique was foundational—targeting the realism of the model's assumptions and the paucity of data for baseline estimation. The authors' defense was technical (convergence stats), which rarely resolves a conceptual disagreement about whether a model reflects reality.

---

### Decision · Program_Chairs · 2026-01-26

Reject